# Re-evaluating malarial retinopathy to improve its diagnostic accuracy in paediatric cerebral malaria: A retrospective study

Kyle J. Wilson[1,2]*, Alice Muiruri Liomba[3], Karl B. Seydel[3,4], Owen K. Banda[2], Christopher A. Moxon[2,5], Ian J. C. MacCormick[6,7], Simon P. Harding[1,8], Nicholas A. V. Beare[1,8], Terrie E. Taylor[3,4]

**1** Department of Eye and Vision Science, University of Liverpool, Liverpool, United Kingdom, **2** Malawi-Liverpool-Wellcome Programme, Blantyre, Malawi, **3** Blantyre Malaria Project, Kamuzu University of Health Sciences, Blantyre, Malawi, **4** College of Osteopathic Medicine, Michigan State University, East Lansing, Michigan, United States of America, **5** School of Infection and Immunity, University of Glasgow, Glasgow, United Kingdom, **6** Institute for Neuroscience and Cardiovascular Research, University of Edinburgh, Edinburgh, United Kingdom, **7** Institute for Adaptive and Neural Computation, College of Science & Engineering, University of Edinburgh, Edinburgh, United Kingdom, **8** St Paul's Eye Unit, Liverpool University Hospitals Foundation NHS Trust, Liverpool, United Kingdom

* kyle.wilson@liverpool.ac.uk

## Abstract

### Background

Previous work has identified that malarial retinopathy has diagnostic value in paediatric cerebral malaria (CM). To improve our understanding of malarial retinopathy as a predictor of cerebral parasite sequestration in paediatric CM we reviewed data from the Blantyre autopsy study, to test the hypothesis that malarial retinopathy is an accurate predictor of cerebral parasite sequestration in an autopsy cohort.

### Methods and findings

We performed a retrospective analysis of data collected from a consecutive series of patients presenting to the Pediatric Research Ward at Queen Elizabeth Central Hospital in Blantyre, Malawi between 1996 and 2010. We determined the diagnostic accuracy of malarial retinopathy as a predictor of cerebral parasite sequestration in a cohort of children with fatal CM. Of 84 children included in the study, 65 met the World Health Organization clinical diagnostic criteria for CM during life. Eighteen (28%) of 65 did not have evidence of cerebral parasite sequestration at autopsy and 17 had an alternative cause of death. Malarial retinopathy had a sensitivity of 89.4% (95% CI [77.6%, 95.6%]) and specificity of 73.0% (95% CI [57.2%, 84.8%]) to predict cerebral parasite sequestration. In a subset of patients with graded retinal assessments, this was improved to 94.3% (95% CI [81.7%, 98.7%]) and 88.0% (95% CI [70.4%, 96.2%]) by reclassifying patients in whom the only retinal sign was 1–5

**Data availability statement:** Data required to reproduce this analysis have been provided along with this manuscript (see Supporting information). The code used in the analysis is available from Github [https://github.com/Kajlinko/ReevaluatingRetinopathy/tree/v1.1.0] and archived in Zenodo [https://doi.org/10.5281/zenodo.17063108].

**Funding:** This work was supported by Wellcome [223502 (K.J.W.) & 222530 (N.A.V.B.)] (https://wellcome.org/). The funders played no role in the study design, data collection and analysis, decision to publish, or preparation of the manuscript.

**Competing interests:** The authors have declared that no competing interests exist.

**Abbreviations:** CM, cerebral malaria; LASSO, least absolute shrinkage and selection operator; STARD, Standards of Reporting Diagnostic Accuracy.

haemorrhages in a single eye as retinopathy negative. This study is limited by its retrospective nature and the inherent selection bias associated with autopsy studies.

## Conclusions

Malarial retinopathy remains the most specific point-of-care test for CM in endemic areas. Its specificity may be improved, without sacrificing sensitivity, by reclassifying patients in whom the only retinal sign is fewer than 5 haemorrhages in a single eye as malarial retinopathy negative. A management algorithm is proposed for integration of malarial retinopathy into clinical care in both well-resourced and resource-limited environments.

---

## Author summary

### Why was this study done?

- In countries where malaria is common, it is possible that a child with malaria parasites in their blood and coma from a nonmalarial cause could be misdiagnosed as cerebral malaria (CM).

- Malarial retinopathy, a collection of features seen in the retinas of children with severe malaria, can help distinguish true CM from coma of other cause.

- However, the only study which showed this previously had a small sample size and didn't report the alternative causes of death.

### What did the researchers do and find?

- Using updated data from the original autopsy study, which lasted 14 years, we confirmed that malarial retinopathy accurately predicts true CM in a cohort of children who died.

- We examined all the cases where malarial retinopathy failed to correctly predict true CM and suggest an updated definition of malarial retinopathy which may improve the quality of predictions.

- We also report the alternative causes of death for all children with malaria who died without evidence of CM.

### What do these findings mean?

- These findings help clinicians to identify comatose children who are at risk of misdiagnosis and identify the lungs, liver, and brain as key targets for investigation into alternative diagnoses.

- This study uses autopsy data, so it may not be generalisable to cohorts which include survivors and future work should seek to address the challenge of definitive CM diagnosis during life.

## Background

The burden of fatal cerebral malaria (CM) continues to be high despite decades of high-quality research. In 2022, malaria claimed over 600,000 lives, most being African children aged 0–5 years [1]. Unlike African adults, young children in malaria-endemic regions of sub-Saharan Africa have not yet developed partially protective immunity, leaving them particularly vulnerable to CM, a severe neurological manifestation of the disease [2]. CM has high mortality even with treatment, and survivors very commonly suffer from neurological injury [3]. CM also affects adults, but different pathophysiological mechanisms may underpin the disease [4]. Here, our analysis concerns Malawian children and, unless otherwise stated, references to CM refer specifically to paediatric disease. In children, CM is defined clinically by the World Health Organization (WHO) as coma (Blantyre Coma Score ≤ 2) with asexual parasitaemia on a blood film and in the absence of alternative or transient causes of coma [5].

Malarial retinopathy has diagnostic and prognostic value in CM [6,7]. In a landmark autopsy study, Taylor and colleagues established the reference standard for definitive CM diagnosis as histopathological evidence of cerebral parasite sequestration in the cerebral microvasculature and malarial retinopathy as the only clinical feature which differentiated histopathologically-proven CM from coma of other cause [6]. In this autopsy cohort (from 1996–2000), 23% of cases defined as CM by WHO clinical diagnostic criteria had no or minimal cerebral parasite sequestration and other causes of death, suggesting misclassification. When malarial retinopathy, detected by indirect ophthalmoscopy, was added to the definition of CM, the sensitivity and specificity for predicting cerebral parasite sequestration improved to 95% and 90%, respectively [8].

An additional 61 autopsies were performed following the initial analyses in 2004. Here, we provide complete results from the largest autopsy study conducted on paediatric CM cases, re-evaluating the hypothesis that malarial retinopathy accurately predicts cerebral parasite sequestration in an autopsy cohort. We consider these results alongside work published in the two decades following the publication of the findings from the initial autopsy cohort to provide updated estimates of the diagnostic accuracy of malarial retinopathy.

## Methods

### Ethics statement

Ethical approval was granted by the research ethics committees at Michigan State University (00-836) and the University of Malawi College of Medicine (P.99/00/100 and P.11/07/593), and research was conducted in accordance with the tenets of the Declaration of Helsinki. Written informed consent for participation was obtained from parents or guardians of all participants.

**Study design.** We conducted a retrospective analysis of autopsy cases from the Blantyre Autopsy Study, a case-control study conducted in Queen Elizabeth Central Hospital, Blantyre, Malawi, between 1996 and 2010. We extracted information on clinical diagnosis, histopathological diagnosis, retinopathy status, and cause of death. We then established estimates for the performance of malarial retinopathy as a predictor of cerebral parasite sequestration, the pathological hallmark of CM. This study has been reported in accordance with the Standards of Reporting Diagnostic Accuracy (STARD) guideline (S1 Checklist).

**Participants.** Our cohort was drawn from a consecutive series of patients in coma admitted to the Pediatric Research Ward. In the event of a death, and when the parents or guardians consented, individuals were included. Clinical diagnoses were determined prior to each autopsy according to the following criteria:

*Cerebral malaria—WHO definition*

- Admitting Blantyre Coma Score ≤ 2, remaining ≤2 following correction of hypoglycaemia, for 30 min after the cessation of seizure activity, or 2 hours from admission, and

- *Plasmodium falciparum* parasitemia, and

- No evidence of meningitis following lumbar puncture (i.e., <10 white blood cells/µL, no pathogens seen or cultured).

*Nonmalarial coma*

- Admitting Blantyre Coma Score ≤ 2, remaining ≤2 following correction of hypoglycaemia, for 30 min after the cessation of seizure activity, or 2 hours from admission, and

- No evidence of malaria infection on a maximum of 4 blood films collected every 6 hours, or

- Nonmalarial aetiology of coma identified during life, irrespective of peripheral parasitemia.

Detailed descriptions of the patients have been published elsewhere [6].

**Test methods.** *Index test—presence of malarial retinopathy:* Patients underwent retinal examinations on admission by direct and indirect ophthalmoscopy after both eyes were dilated with 1% tropicamide and 2.5% phenylephrine eye drops. Where dilation was insufficient, the drops were repeated. All examinations were performed by either an ophthalmologist or a clinician trained in retinal examination. Ophthalmologists graded malarial retinopathy findings according to a validated grading system and recorded them on a standardised form [9]. Representative images of graded retinal findings are provided by Harding and colleagues [9]. Trained clinicians recorded retinopathy status as either present or absent. Test positivity for clinicians was defined as the presence of any retinal haemorrhages, retinal whitening, or retinal vessel change in either eye. When considering graded ophthalmologist assessments, test positivity was defined as presence of any retinal haemorrhages, retinal whitening (at the macula, fovea, or in any quadrant of the periphery), or retinal vessel change (in any quadrant of the periphery) in either eye. Test negativity was defined as negative for retinal haemorrhages, macular and foveal whitening, and negative peripheral whitening and vessel change with a minimum of three out of four quadrants visualised in both eyes. Clinicians performing fundoscopy had access to clinical information about the patient. All fundoscopies were performed during life.

*Reference standard—histopathological evidence of cerebral parasite sequestration:* The autopsy procedure has been described in detail elsewhere [6]. Briefly, in the event of death consent was sought for autopsy by a member of the clinical team, and if granted, performed as soon as possible in the hospital mortuary.

Tissue blocks from standard regions of the cerebrum, brainstem, and cerebellum were fixed in 10% neutral buffered formalin, processed, and embedded in paraffin. Haematoxylin and eosin were used to stain 3–5 µm sections. The contents of at least one hundred perpendicularly cross-sectioned capillaries per section were counted by a single observer. Any blood vessel, circular or oval in profile, with a maximum:minimum diameter of <2:1, and with at most one visible endothelial cell nucleus in the wall, was considered to be a capillary.

Test positivity was defined as presence of sequestered parasitised red blood cells in the cerebral microvasculature, as determined by a histopathologist. This corresponded to a cut-off value of 23.3% parasitised capillaries, as determined by classification and regression trees [6]. The pathologist performing histopathological assessment was masked to clinical information and retinopathy findings.

**Analysis.** All analyses were conducted in RStudio using R 4.4.0. Malarial retinopathy was treated as a binary diagnostic test to predict cerebral parasite sequestration. Sensitivity, specificity, positive predictive value, negative predictive value, positive likelihood ratio, and negative likelihood ratio were calculated using the *testCompareR* package, which utilises standard formulae. Following testing of the primary hypothesis, we generated a new hypothesis based upon the data—that the predictive value of malarial retinopathy could be improved by slightly altering the definition. Statistical inference to compare the performance metrics of both definitions of malarial retinopathy (diagnostic tests) was also handled by *testCompareR*. We have comprehensively documented the methods employed by *testCompareR* for inference elsewhere [10]. *p* values were adjusted using the Holm method to correct for multiple comparisons. Findings were considered significant if the adjusted *p* value was <0.05.

Tests were either considered positive, negative, or missing. Missing data were handled by complete case analysis and validated using extreme case analysis. Because we had <5% missing data in the reference test, we excluded the four cases with missing reference test data and performed sensitivity analysis in the 15 remaining cases of missing index test data.

To evaluate the effects of clinical parameters on results of the index and reference tests, we used a logistic regression model with the least absolute shrinkage and selection operator (LASSO) to predict the outcome of the reference test using the index test result, a selection of relevant clinical parameters and their interactions with the index test [11]. Where meta-data was missing, we used multiple imputation by chained equations to impute missing data [12]. The number of times a variable was selected by the LASSO model across imputations was used as a simple heuristic method of determining variable importance.

Sample size was dictated by the retrospective nature of this study.

## Results

Sixty-one autopsies were performed following the initial analysis in 2004, making a total of 103 autopsy examinations (Fig 1). There were 84 cases with retinal data. Sixty-five of these met WHO criteria for CM during life. Patient demographics are described in Table 1. Nineteen cases had missing data for either the index or reference test. The exact reasons for missing data were not recorded at the time. These may include the participant being too unstable to undergo fundoscopy, no examiner skilled in fundoscopy being available or loss of data between the time of the examination and the time of this analysis.

First, we sought to evaluate the diagnostic accuracy of malarial retinopathy as a predictor of cerebral parasite sequestration to validate the findings from the previous study. We found a sensitivity of 89.4%, which is similar to the initial analysis [8]. However, the specificity was 73.0%, considerably lower than the initial analysis.

Because changes in specificity are due to false positives, we reviewed the case notes for all false positives to identify the nature of their retinopathy. In four of 10 false positives, the only evidence of malarial retinopathy was 1–5 haemorrhages in a single eye. The retinal phenotype for all 10 of these patients is described in S1 Analysis. Based on this chart review, we hypothesised that the current definition of malarial retinopathy as a predictor of cerebral parasite sequestration could be improved. Specifically, if the only retinal finding is 1–5 haemorrhages in a single eye, this may be insufficient to predict cerebral parasite sequestration. Generation of this new hypothesis was data-driven.

A total of 64 patients had graded retinal exams. However, a total of four ophthalmologist assessments were deemed to be ungradable based upon our index test criteria. To test our hypothesis, we reanalysed the 60 patients in whom a graded retinal examination was performed by an ophthalmologist according to the validated system [8]. Where the only evidence of malarial retinopathy was 1–5 haemorrhages in a single eye, the data were re-coded as no longer constituting a diagnosis of malarial retinopathy. Contingency tables for all analyses are shown in Table 2. The test metrics were compared for the original and the re-coded data (Table 3).

Re-categorising the participants reduced the number of false positives without a corresponding increase in false negatives, resulting in an improvement in the specificity of malarial retinopathy of 12% (95% CI [−4.5%, 26.8%]; $p = 0.065$).

Worst- and best-case analysis for the missing data returned point estimates that fell within the confidence intervals established for sensitivity and specificity during complete case analysis. In the worst-case scenario specificity, positive predictive value and positive likelihood ratio fell to 58.7%, 68.9%, and 1.9, respectively. Sensitivity, negative predictive value, and negative likelihood ratio remained moderate to good at 79.2%, 71.1%, and 0.4, respectively (S2 Analysis). Missing patients tended to be older (and therefore heavier), with lower cerebrospinal fluid opening pressure and shorter time to death than included patients (S1 Table). However, these parameters were suppressed to zero by the LASSO regularisation method in a logistic regression model to predict cerebral parasite sequestration from the malarial retinopathy status and clinical parameters, suggesting they exert little influence on the results of the index or reference tests (S2 Analysis).

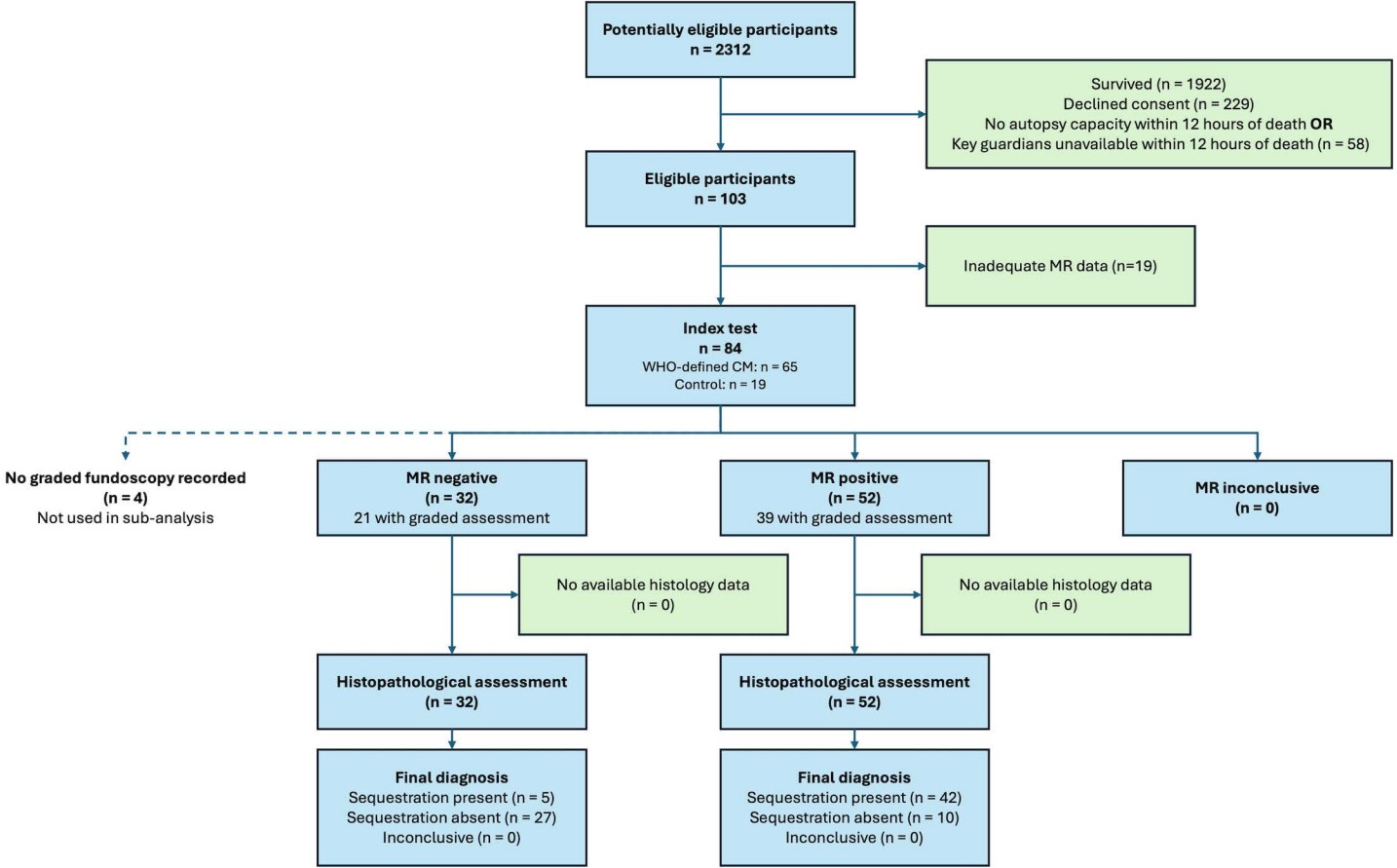

**Fig 1. STARD flow chart to describe flow of data through the study.** Abbreviations: CM, cerebral malaria; MR, malarial retinopathy; WHO, World Health Organization.

Eighteen of 65 children (28%) met the clinical case definition for CM but had no evidence of cerebral parasite sequestration at autopsy. An alternative cause of death was evident in 17 (Table 4). Four (22%) of these cases had evidence of retinopathy using the original definition, falling to 3 (17%) using the updated definition.

## Discussion

This retrospective study confirms that a notable proportion of patients apparently dying of WHO-defined CM have a nonmalarial cause of death. Identifying children who have an alternative cause for their coma is therefore critical. The high negative predictive value (84.4%) and low negative likelihood ratio (0.1) observed in our cohort suggests that absence of malarial retinopathy is an effective way to identify these children. Interestingly, specificity was lower than seen in the only other study of CM autopsy cases to include malarial retinopathy findings [6]. This was due to several false positives in which the only retinal sign was a small number of haemorrhages in a single eye. Reclassifying patients in whom the only retinal sign of malaria is 1–5 haemorrhages in a single eye may improve the specificity of malarial retinopathy as a diagnostic test. This compelling trend further corroborates recent work by our group which showed that retinal haemorrhages alone are insufficient to accurately predict cerebral parasite sequestration in the autopsy cohort, but that retinal whitening and retinal vessel change perform well as predictors [13]. It should be noted that the inherent limitations of retrospective research mean that we

**Table 1. Demographics summarised by the presence of cerebral parasite sequestration and the presence of malarial retinopathy.**

| Cerebral parasite sequestration | All | − | − | + | + |
|---|---|---|---|---|---|
| Malarial retinopathy | All | − | + | − | + |
| | $n=84$ | $n=27$ | $n=10$ | $n=5$ | $n=42$ |
| Age | 29 (6–127) | 29 (7–108) | 20 (6–96) | 57.5 (14–103) | 30 (6–127) |
| No. male (%)* | 46 (54.8) | 16 (59.3) | 6 (60) | 3 (60) | 21 (50) |
| No. fever (%)* | 71 (85.5) | 18 (69.2) | 8 (80) | 4 (80) | 41 (97.6) |
| Fever duration (h) | 48 (2–504) | 48 (2–504) | 36 (4–72) | 72 (24–216) | 48 (4–120) |
| Temperature (°C) | 38 (32.2–42.1) | 37.5 (32.2–42.1) | 37.6 (35–40.1) | 37.8 (33.9–40.3) | 38.8 (36.5–41.9) |
| Heart rate (/min) | 150 (52–233) | 144 (92–180) | 152 (52–180) | 144 (120–200) | 160 (60–233) |
| Respiratory rate (/min) | 46 (20–84) | 44 (32–68) | 42 (20–66) | 48 (28–50) | 48 (28–84) |
| Systolic BP (mmHg) | 104 (70–175) | 100 (74–160) | 110 (90–175) | 110 (94–110) | 102 (70–140) |
| Weight (kg) | 10.4 (4–31) | 10.5 (4–26) | 10 (6.5–20.5) | 19 (9.1–31) | 10 (6.8–27) |
| Haematocrit (%) | 22.6 (4–49) | 30 (4–42) | 27 (7–49) | 21 (17–35) | 18 (10–35) |
| No. anaemia (%)* | 19 (22.6) | 6 (22.2) | 3 (30) | 0 (0) | 10 (23.8) |
| No. BCS 0 (%)* | 30 (35.7) | 14 (51.9) | 3 (30) | 2 (40) | 11 (26.2) |
| No. BCS 1 (%)* | 33 (39.3) | 6 (22.2) | 5 (50) | 2 (40) | 20 (47.6) |
| No. BCS 2 (%)* | 17 (20.2) | 4 (14.8) | 1 (10) | 1 (20) | 11 (26.2) |
| No. BCS>2 (%)* | 4 (4.8) | 3 (11.1) | 1 (10) | 0 (0) | 0 (0) |
| CSF opening pressure (mmH$_2$O) | 190 (9–320) | 215 (20–320) | 205.5 (55–320) | 170 (80–230) | 190 (9–320) |
| Time to death (h) | 12.5 (0–168) | 14 (0–168) | 4 (1–64) | 18 (5–56) | 12.5 (1–46) |

*indicates the number of participants meeting specified condition (percentage). All other values indicate the median (range). Abbreviations: BCS, Blantyre Coma Score; BP, blood pressure; bpm, beats per minute; CSF, cerebrospinal fluid; rpm, respirations per minute.

**Table 2. Contingency tables.**

| All data ($n=84$) | Malarial retinopathy | |
|---|---|---|
| Cerebral sequestration | + | − |
| + | 42 | 5 |
| − | 10 | 27 |
| Ophthalmologist-graded ($n=60$) | Malarial retinopathy | |
| Cerebral sequestration | + | − |
| + | 33 | 2 |
| − | 6 | 19 |
| Re-coded ($n=60$) | Malarial retinopathy | |
| Cerebral sequestration | + | − |
| + | 33 | 2 |
| − | 3 | 22 |

Retinal findings for each of the analyses (whole dataset; ophthalmologist-graded; re-coding 1–5 haemorrhages in a single eye as malarial retinopathy negative) compared against the gold standard, cerebral parasite sequestration.

were not adequately powered to detect a difference in specificity ($P=0.065$) nor positive predictive value or likelihood ratio (both $P=0.34$ after Holm correction). To definitively validate this finding a larger autopsy study, or a model which very reliably predicts cerebral parasite sequestration from clinical parameters measured during life, is needed. Current models are either insufficiently sensitive to be used as a reference test [14], or they include retinopathy as a predictor [15].

**Table 3. Performance metrics for malarial retinopathy as a predictor of cerebral parasite sequestration.**

| Performance metric | % (95% Confidence Interval) | | | |
| --- | --- | --- | --- | --- |
| | All data (*n*=84) | Ophthalmologist-graded (*n*=60) | Re-coded (*n*=60) | *\*p* |
| Sensitivity | 89.4 (77.6, 95.6) | 94.3 (81.7, 98.7) | 94.3 (81.7, 98.7) | – |
| Specificity | 73.0 (57.2, 84.8) | 76.0 (56.8, 88.8) | 88.0 (70.4, 96.2) | 0.065 |
| Positive predictive value | 80.8 (71.2, 87.9) | 84.6 (70.5, 93.0) | 91.7 (78.5, 97.4) | 0.34 |
| Negative predictive value | 84.4 (75.2, 90.7) | 90.5 (71.5, 97.8) | 91.7 (74.6, 98.1) | 0.34 |
| Positive predictive value | 3.3 (2.0, 5.7) | 3.9 (2.1, 7.7) | 7.9 (3.1, 19.0) | 0.34 |
| Negative predictive value | 0.1 (0.1, 0.3) | 0.1 (0.1, 0.2) | 0.1 (0.0, 0.2) | 0.34 |

All data includes physician assessment of presence of malarial retinopathy; ophthalmologists performed a complete grading. *\*p* values compare the cohort with graded retinopathy assessments before and after re-coding 1–5 haemorrhages in a single eye as insufficient to define malarial retinopathy using the testCompareR package in R. All *p* values have been adjusted for multiple comparisons using the Holm method.

**Table 4. Causes of death and retinopathy status in children meeting the clinical definition of cerebral malaria during life but without cerebral parasite sequestration at autopsy.**

| Patient | Cause of death | Malarial retinopathy status | |
| --- | --- | --- | --- |
| | | Old definition | New definition |
| 1 | Hepatic necrosis | – | – |
| 2 | Reye's syndrome, *Streptococcus pneumoniae* pneumonia | – | – |
| 3 | Reye's syndrome, viral pneumonia | – | – |
| 4 | *S. pneumoniae* pneumonia | – | – |
| 5 | Reye's syndrome, viral pneumonia | – | – |
| 6 | Ruptured arteriovenous malformation | + | + |
| 7 | Anaemia | + | + |
| 8 | *Klebsiella oxytoca* sepsis with secondary meningitis | – | – |
| 9 | Skull fractures | + | + |
| 10 | Hepatitis | – | – |
| 11 | *S. pneumoniae* pneumonia | – | – |
| 12 | Subdural and intracerebral haematomas | + | – |
| 13 | Pneumonia, organism unknown | – | – |
| 14 | Pneumonia, meningoencephalitis | – | – |
| 15 | Pneumonia with secondary meningitis | – | – |
| 16 | Left ventricular failure with pulmonary oedema | – | – |
| 17 | No alternative cause of death identified | – | – |
| 18 | Pneumonia | – | – |

A major strength of autopsy studies is that they can provide a reference standard by which clinical diagnoses can be validated. However, autopsy studies have inherent selection bias for fatal cases which may compromise the generalizability of their results. This highlights an important question: if surviving malarial retinopathy negative children do not have CM, what do they have?

A large comparative study found that children with malarial retinopathy presented later in their illness, had longer hospital admissions and coma duration and had higher parasite biomass than those without retinopathy [16]. Rates of bacteraemia did not differ. These findings suggest that CM represents a spectrum of disease, with malarial retinopathy-negative CM falling at the less severe end. Subsequent analyses have demonstrated increased inflammation in malarial retinopathy positive versus malarial retinopathy negative CM [17], and *var* genes associated with severe malaria are most highly

expressed in malarial retinopathy positive CM, followed by malarial retinopathy negative CM, severe malarial anaemia, and asymptomatic parasitaemia [18]. Furthermore, children with malarial retinopathy negative CM are more likely to have pre-existing developmental problems and family history of epilepsy, implying increased susceptibility to coma following malaria infection [19]. Modelling studies which include survivors estimate that only 7%–11% of comatose, parasitemic children have a nonmalarial aetiology for their disease [15,20].

Together with our autopsy findings, these data suggest that malarial retinopathy-negative CM patients are a heterogenous group, containing patients for whom the major contributing pathogenetic mechanism of their coma is malarial parasitemia. These patients may have a less severe clinical manifestation of CM or have a predisposition to coma. The malarial retinopathy negative CM group also contains children who have a diverse range of nonmalarial pathologies (see Table 2). This group ranges from 10% (in children admitted to hospital with WHO-defined CM) to 28% (in fatal cases of WHO-defined CM), justifying comprehensive investigation for other causes of coma in this group and following initiation of treatment for CM. Specifically, those without malarial retinopathy who fail to improve on treatment would benefit from investigations which target the liver, lungs, and brain. These findings have important implications for allocation of resources in settings where access may be limited. While the introduction of fundoscopy into routine clinical practice could be hampered by the relative paucity of ophthalmic specialists in malaria-endemic areas [21], we have recently shown that structured simulation training can rapidly equip nonophthalmic specialists to perform fundoscopy for malarial retinopathy using low-cost instruments [22]. A management pathway which considers resource levels and the currently available evidence on malarial retinopathy is proposed in Fig 2.

There are also important implications for research. Including patients without CM in trials of adjunctive therapies for CM hamper efforts to develop adjunctive treatments. Studies could inadvertently be underpowered to detect a difference between interventional groups if they include non-CM patients. Equally, results in mechanistic studies could be substantially altered by the including patients without CM. Further work is required to definitively characterise malarial retinopathy negative CM, quantifying the proportion of children who have an alternative cause for coma and identifying the factors increasing the susceptibility to coma in some children with malaria infections. These factors may be altered as malaria vaccine programs are rolled out on larger scales.

This study does have limitations. As previously discussed, autopsy cases carry an inherent selection bias and, though this has been addressed elsewhere in the discussion, the results may not be generalisable to cohorts which include survivors. Furthermore, of the 390 patients who died during the study period, only 103 underwent autopsy. Several factors can influence a guardian's decision to grant consent for autopsy and, while this may be random, it is impossible to exclude systematic differences that could introduce bias in the data.

We were unable to compare the original cohort of 42 patients to the more recent cohort of 61 patients as, 20 years after the original, some data were irretrievable. As a result, our analysis reflects the pooled results of both cohorts. The adjusted definition of malarial retinopathy was determined post-hoc after careful chart review and a single hypothesis was formed and tested. However, because no validation data set exists the hypothesis was derived and tested in the same data set. This highlights an important gap for future research into malarial retinopathy or other diagnostic tests for cerebral malaria (CM). Minimally invasive tissue sampling likely represents the most appropriate method for getting a definitive post-mortem diagnosis and could be employed across multiple sites [23].

Complete blinding to clinical condition was not possible, as examining clinicians were directly involved in patient care. It has been shown previously that fundoscopy has good inter-rater reliability in this clinical setting, supporting the validity of this method [24]. Alternatives, such as grading of retinal photos, were not routinely available in Malawi for much of the study period. Furthermore, retinal photography provides an inadequate field of view, resulting in systematic under-grading as shown in S3 Analysis and discussed elsewhere [25]. While this may change in the future, for now, indirect ophthalmoscopy remains the gold standard method for determining malarial retinopathy status.

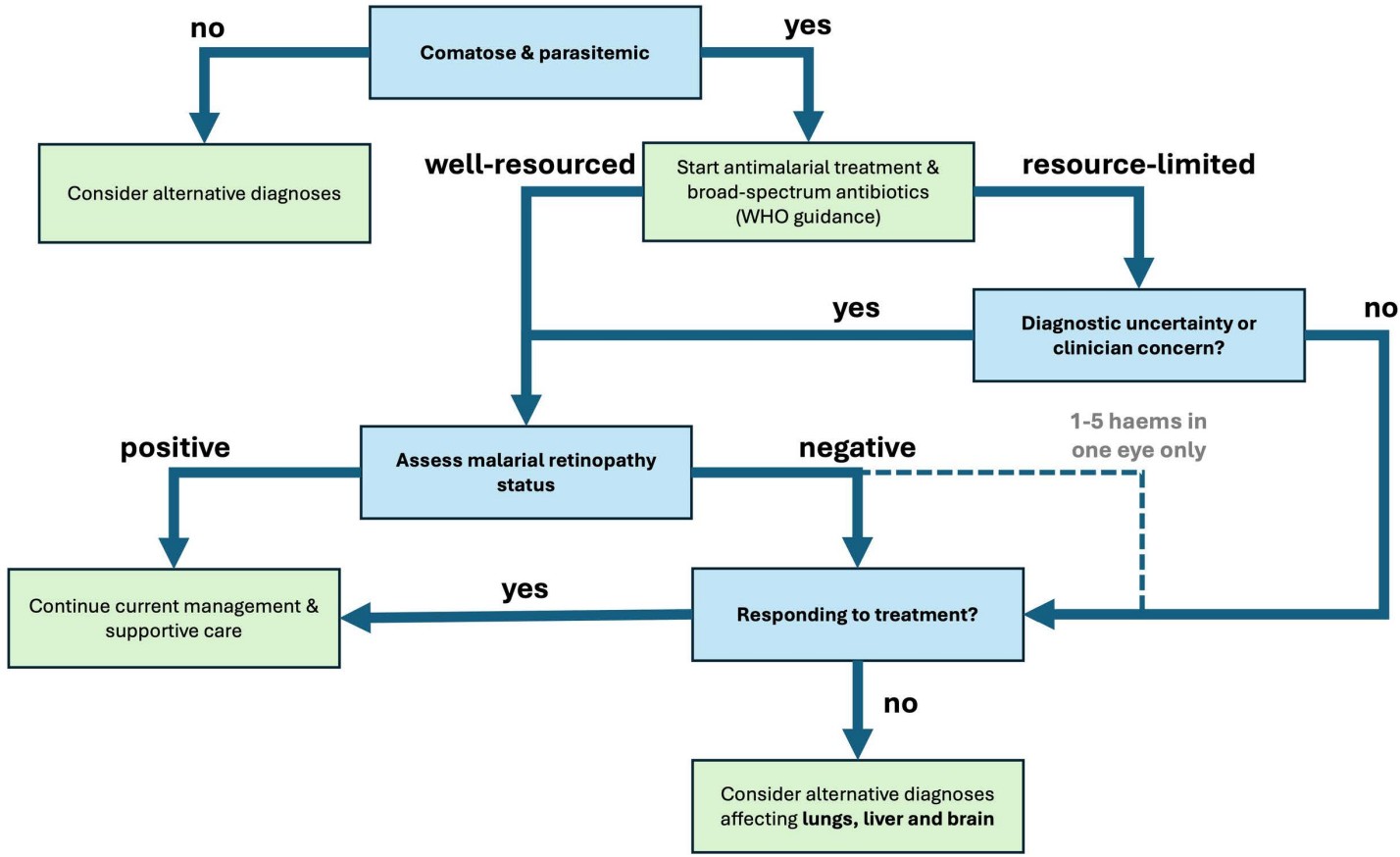

**Fig 2. Management algorithm for comatose, parasitemic children in well-resourced and resource-limited settings.** Abbreviation: WHO, World Health Organization.

Anaemia is common in malaria-endemic settings and may cause anaemic retinopathy [26], potentially leading to false-positive retinal findings in the absence of cerebral parasite sequestration. In our cohort, the difference in anaemia incidence between malarial retinopathy positive and negative children without cerebral sequestration was small, suggesting limited confounding; nonetheless, this warrants further investigation. Among children with cerebral parasite sequestration, we observed a more pronounced difference in anaemia incidence between malarial retinopathy groups, albeit with a small sample size in the retinopathy negative group. Disentangling the retinal effects of anaemia from those of cerebral parasite sequestration is challenging. While recent studies have highlighted the co-existence of multiple pathologies in febrile coma, including contributions from CM [27], they have not incorporated detailed retinal phenotyping—an important gap for future work aiming to clarify the interplay between systemic and retinal pathologies.

Finally, because granular details of the retinal presentation are missing from patients assessed by generalists, the sub-analysis to assess the effect of re-defining the index test as negative when only 1–5 haemorrhages are present in a single eye was necessarily performed using ophthalmologist-graded assessments. In our experience, the inter-rater reliability between ophthalmologists and trained clinicians in detecting malarial retinopathy is good. Certainly, clinicians with different levels of experience grading images of malarial retinopathy have good inter-rater reliability (S3 Analysis). Nevertheless, using these data alone it is perhaps more correct to say that we assessed the combined effect of re-defining the index test as negative when only 1–5 haemorrhages are present in a single eye and restricting examinations to those

performed by ophthalmologists. Going forward, in situations where ophthalmologist examination is impractical, such as large multi-site trials, AI-assisted analysis of images acquired at the bedside may soon be a feasible alternative, provided that the technologies permit good visualisation of the retinal periphery [25].

At present, malarial retinopathy remains the most specific point-of-care test for CM in endemic areas and its specificity may be improved, without sacrificing sensitivity, by reclassifying patients in whom the only retinal sign is fewer than 5 haemorrhages in a single eye as malarial retinopathy negative.

## Supporting information

**S1 Checklist. STARD Checklist.** Standards for reporting of diagnostic accuracy studies checklist.
(DOCX)

**S1 Analysis. Quarto document containing primary analyses.** Analysis required to produce tables and figures included in the main article, along with annotations explaining the rationale.
(PDF)

**S2 Analysis. Quarto document containing supplementary analyses.** Analysis required to produce tables and figures included in the supplementary information. Contains sensitivity analyses and analyses of spectrum bias.
(PDF)

**S3 Analysis. Quarto document containing supplementary analyses.** Analysis required to verify inter-rater reliability of grading ophthalmic images.
(PDF)

**S1 Table. Description of participants included in complete case analysis and those excluded due to missing reference or index test results.**
(DOCX)

**S1 Data. Anonymised data files required to reproduce the analysis - original autopsy data.**
(XLSX)

**S2 Data. Anonymised data files required to reproduce the analysis - chart review data.**
(XLSX)

**S3 Data. Anonymised data files required to reproduce the analysis - graded retinopathy data.**
(CSV)

**S4 Data. Anonymised data files required to reproduce the analysis - clinical and demographic data.**
(XLSX)

**S5 Data. Anonymised data files required to reproduce the analysis - retinopathy image grading data.**
(XLSX)

**S6 Data. Anonymised data files required to reproduce the analysis - graded retinopathy data for imaging cases.**
(CSV)

## Acknowledgments

We would like to thank the parents and guardians of the children included in this study for their selfless gift, as well as the staff on the Pediatric Research Ward and in the mortuary of Queen Elizabeth Central Hospital, without whom this study would not have been possible.

## Author contributions

**Conceptualization:** Kyle J. Wilson, Alice Muiruri Liomba, Karl B. Seydel, Nicholas A. V. Beare, Terrie E. Taylor.

**Data curation:** Kyle J. Wilson, Alice Muiruri Liomba, Karl B. Seydel, Owen K. Banda.

**Formal analysis:** Kyle J. Wilson.

**Funding acquisition:** Kyle J. Wilson, Nicholas A. V. Beare.

**Methodology:** Kyle J. Wilson, Karl B. Seydel, Ian J. C. MacCormick, Terrie E. Taylor.

**Supervision:** Karl B. Seydel, Christopher A. Moxon, Ian J. C. MacCormick, Nicholas A. V. Beare, Terrie E. Taylor.

**Validation:** Owen K. Banda.

**Writing – original draft:** Kyle J. Wilson.

**Writing – review & editing:** Kyle J. Wilson, Alice Muiruri Liomba, Karl B. Seydel, Owen K. Banda, Christopher A. Moxon, Ian J. C. MacCormick, Simon P. Harding, Nicholas A. V. Beare, Terrie E. Taylor.

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
