## [Editor Report · Decision Letter 0]

9 Apr 2025

Dear Dr Wilson, 

Thank you for submitting your manuscript entitled "Re-evaluating malarial retinopathy to improve its diagnostic accuracy in cerebral malaria" for consideration by PLOS Medicine.

Your manuscript has now been evaluated by the PLOS Medicine editorial staff as well as by an academic editor with relevant expertise and I am writing to let you know that we would like to send your submission out for external peer review.

Please re-submit your manuscript within two working days, i.e. by Apr 11 2025 11:59PM.

Kind regards,

Suzanne De Bruijn, PhD

Senior Editor

PLOS Medicine

---

## [Decision Letter · Decision Letter 1]

19 May 2025

Dear Dr Wilson,

Many thanks for submitting your manuscript "Re-evaluating malarial retinopathy to improve its diagnostic accuracy in cerebral malaria" (PMEDICINE-D-25-01141R1) to PLOS Medicine. The paper has been reviewed by subject experts and a statistician; their comments are included below and can also be accessed here: [LINK]

As you will see, the reviewers thought your study is interesting, but also have several concerns. After discussing the paper with the editorial team and an academic editor with relevant expertise, I'm pleased to invite you to revise the paper in response to the reviewers' comments. We like you to address all the reviewer's concerns. We plan to send the revised paper to some or all of the original reviewers, and we cannot provide any guarantees at this stage regarding publication.

We ask that you submit your revision by Jun 09 2025 11:59PM. However, if this deadline is not feasible, please contact me by email, and we can discuss a suitable alternative.

Don't hesitate to contact me directly with any questions (sbruijn@plos.org). 

Best regards, 

Suzanne 

Suzanne De Bruijn, PhD 

Associate Editor

PLOS Medicine

sbruijn@plos.org

Comments from the academic editor:

Please discuss how you would suggest to increase the uptake of retinal examination in busy clinical settings where malaria is endemic, or whether you think that retinoscopy is exclusively a research tool. 

Comments from the reviewers: 

Reviewer #1: This research extends an earlier paediatric autopsy study to further investigate the value of malarial retinopathy (MR) in diagnosing cerebral malaria (CM). A retrospective analysis was performed on autopsy data (the reference standard for diagnosing CM) and clinical diagnostic data for both MR and CM. The MR clinical diagnoses were performed by either ophthalmologists or clinicians trained in retinal examination. The authors found clinical MR diagnoses generally had good sensitivity and specificity to predict CM, however accuracy was improved by using only the ophthalmologist-diagnosed MR data and a higher threshold for MR diagnosis. 

This is a well-written paper that effectively contextualises and establishes the importance of the research presented. While there are some opportunities for clarifying the the background to the study, the methods used, and the results obtained, these parts of the manuscript are broadly cogent and clear. Indeed, I particularly want to commend the authors for including R markdown annotated code, which I found very helpful in understanding their analyses. That said, I have more substantial concerns about the content and structure of the Discussion and suggest a more comprehensive rethink and rewrite.

Detailed feedback on the manuscript is provided below. My primary focus has been methodological, but I have made a few broader points as well. All items are major unless indicated otherwise.

1. Suggest including 95% confidence intervals for estimated statistics (i.e., sensitivity and specificity) in the abstract. 

2. Suggest the authors briefly explain why the focus of this research is paediatric. The authors state that children bear most of the burden of disease of malaria, but is it also that children are particularly at risk of CM? Or something else entirely? 

3. [MINOR] Consider indicating in the title that the population is paediatric. 

4. Suggest avoiding words like "significant" and "significance" unless discussing statistical significance, accompanied by referenced or actual statistical results. Reconsider wording like "Malarial retinopathy (MR) has diagnostic and prognostic significance in CM." and "This study confirms that a significant proportion of patients apparently dying of WHO-defined CM have a non-malarial cause of death...". 

5. If I am understanding correctly "(cerebral) (parasite) sequestration" is effectively interchangeable in the context of this paper with both "histopathologically-proven CM" and "CM positive (by autopsy)" i.e., this study's reference standard. I found this quite confusing on my first read through - particularly in Figure 1 where the "CM" abbreviation is defined but "sequestration" is used instead. Suggest making these definitions clear early and, as much as possible, sticking to one wording that is readily understandable. 

6. Similarly, the term "WHO criteria" is used with no definition. I suspect this means a set of clinical diagnostic criteria defined by the World Health Organization, but I suggest making this much clearer up front with an appropriate reference. I also note that clinical diagnostic criteria are presented in the Methods, but it's not clear if these are the WHO criteria. 

7. Thinking on this a bit more, I'd suggest making the various ways of ascertaining MR (and their interrelationships) much clearer up front. Something like "The reference standard for ascertaining MR is by autopsy… Clinical approaches also exist, such as the WHO standard… Recent research by Taylor et al raised concerns about the WHO standard … They also found incorporating MR improved diagnostic accuracy…". It doesn't need to be a lot of words but would greatly help orient the reader. 

8. "In CM cohorts which include survivors, the proportion of those with non-malarial coma is estimated to be lower at 10%." Not sure of the relevance of this given the reference standard here is by autopsy in deceased patients. Suggest elaborating or omitting. 

9. "There remains a significant proportion of MR negative CM patients for whom parasitemia is a major contributor to coma, perhaps in combination with other factors, including co-infection and preexisting neurological disease" Again, while interesting I am not clear on the relevance of this in the context of this research - the authors are not attempting to ascertain these factors. Suggest elaborating or omitting.

10. The "Test Methods" section of the Methods is excellent. Very precise and clear. 

11. Perhaps the sentence "A total of four ophthalmologist assessments were deemed to be ungradable based upon these criteria." would be more appropriately located in the Results? 

12. The "Statistics" section of the methods is quite brief, with only a reference to the R package used. Suggest providing enough detail on how statistics were calculated to allow replication outside of the R package - especially calculation of 95% confidence intervals and statistical tests used to produce p-values presented. Also important to note multiplicity adjustment strategies like use of Holm method. 

13. Suggest clearly articulating the hypotheses being tested before the Statistics section (and certainly before p-values in the Results).

14. "MR was treated as a binary diagnostic test to predict cerebral parasite sequestration." I understand the authors used the additional information in the ophthalmological assessments to adjust the MR positive/negative demarcation line and, consequently, improved sensitivity and specificity. Did they also consider having more than two levels of MR as the index test?

15. [MINOR] Suggest including the ethics approval reference numbers. 

16. "There were 84 cases with retinal data." This means ~20% of your eligible participants were lost due to inadequate MR data - not a trivial proportion. Did the authors consider sensitivity analyses or other missing data approaches outside of complete case analyses? Either way, this limitation - and any potential biases arising - should be addressed in the Discussion.

17. Suggest including an "All patients" column in Table 1.

18. Based on Figure 1, of the 390 patients who did not survive only 103 were included in the study. I think it is crucial to address this limitation - and any potential biases arising - in the Discussion. 

19. Also on Figure 1, how are the four ungradable ophthalmologist assessments incorporated into the diagram? 

20. Currently there are elements of both methods and results in the Results on page six. I appreciate the authors are accurately presenting the sequence of events as the research progressed, but they could also consider moving some of methodological description into the Methods. While this comment is venturing into the "stylistic", more dogmatic readers may take issue with the current wording and I feel it would be remiss to not make this point. 

21. Not sure of the value of including "Null hypothesis: PLR1 = PLR2 & NLR1 = NLR2, see Supplementary Methods." in the notes accompanying Table 3, as it's not readily understandable with reference to the table only. Suggest elaborating or omitting.

22. "These results indicate that MR can accurately predict cerebral sequestration in comatose children, but also indicate that the specificity of MR can be improved by a definition that excludes cases that have only 1-5 hemorrhages in a single eye." Perhaps this sentence would be more at home in the Discussion?

23. The Discussion is quite long compared to the rest of the paper and, in some cases, reads more like a literature review. I would suggest a heavy edit of paragraphs two, three, and four of the Discussion down to a single paragraph that concisely answers the question posed at the beginning of the first paragraph: "if surviving MR negative children do not have CM, what do they have?". 

24. Conversely, the Discussion is lacking a sufficiently comprehensive interpretation and contextualisation of the results presented. How do the sensitivity and specificity results achieved compare to other approaches and studies? Are there accepted standards in this (or a similar) clinical context? PPV and NPV results are presented but never discussed. Are they important in this context? Why/why not? What about the approach to data missingness? Complete case analyses can introduce bias in some contexts. If this is this an issue here, should future research be conducted to investigate? What can be done to improve accuracy of diagnoses and patient outcomes given the presumably resource-constrained environment? I pose these questions not as an exhaustive list or mandatory items to cover, rather I hope they provide useful prompts for the authors to consider. Note these are in addition to the items already raised for inclusion in the Discussion. 

25. "Additionally, we were unable to compare the original cohort to the more recent cohort directly as, 20 years after the original analysis, some data were irretrievable." Apologies, I don't understand what you are saying here. You obviously have information on the original cohort in the earlier publication. Or do you mean you wanted to do a pooled analysis and needed the individual patient data? Suggest clarifying. 

26. [MINOR] "In our experience, the interrater reliability between ophthalmologists and trained clinicians in detecting malarial retinopathy is good." A citation would be helpful here if available. 

Reviewer #2: This manuscript by Wilson and colleagues is well written and of clear relevance to the malaria research and healthcare community. It re-examines the diagnostic accuracy of malarial retinopathy as an indicator of cerebral parasite sequestration in a pediatric cohort of fatal cerebral malaria (CM) cases, and proposes reclassifying patients with 1-5 hemorrhages in only one eye as retinopathy-negative. The study is well designed, the results are clearly presented, and the discussion is strong. I have only a few minor suggestions to improve clarity and impact:

1. Please adjust the language in both the abstract and main text to specify that the diagnostic utility of malarial retinopathy pertains to pediatric CM only. Several studies in adults have shown no consistent association between CM and malarial retinopathy.

2. Given that (i) the prevalence of anemia in Malawi exceeds the East African average (55.56%), (ii) anemia can cause anemic retinopathy, (iii) it frequently co-occurs with CM in children, and (iv) the proposed reclassification did not affect the diagnosis of anemia (Table 4), I recommend adding hemoglobin levels to Table 1. It would also be helpful to indicate how many fatal cases had concurrent anemia and to briefly discuss anemia as a potential, albeit likely infrequent, confounding factor in the limitations section.

3. Considering this group's prior work and the common use of "MR" to refer to magnetic resonance, I found the use of the acronym for malarial retinopathy potentially confusing. Consider using the full term or a more distinct abbreviation to avoid ambiguity.

Reviewer #3: The authors conducted a valuable retrospective analysis of 84 paediatric autopsy cases from the Blantyre cohort (1996-2010; page 3) to benchmark the diagnostic accuracy of malarial retinopathy (MR) against histopathologically confirmed cerebral sequestration. Among 65 WHO‐defined cerebral malaria (CM) cases, the standard MR assessment achieved sensitivity 89.4% (95% CI 77.6-95.6) and specificity 73.0% (57.2-84.8) (Table 3, page 7). By reclassifying patients with only 1-5 unilateral haemorrhages as MR‐negative, the authors report a rise in specificity to 88.0% (70.4-96.2) while sensitivity improved to 94.3% (81.7-98.7) (Table 3). This simple adjustment, if robust (see comments on methodology below), could markedly reduce misclassification of non‐CM cases and improve patient triage in resource‐limited settings.

Strengths

1. Gold‐standard reference. The use of histologically confirmed brain sequestration in 65 CM cases (page 6, Fig 1) ensures that sensitivity and specificity estimates rest on the true biological gold standard rather than imperfect clinical surrogates. This lends rare and undeniable rigor to the findings.

2. Immediate clinical relevance. A single, low‐cost rule change (1-5 haemorrhages → MR-negative) yields a clinically meaningful 12% gain in specificity without sacrificing sensitivity. In high‐burden clinics, this precision could translate into hundreds of accurately diagnosed children each year.

3. Detailed data validation. Supplementary Methods (pages 21-32) describe exhaustive chart reviews and database audits culminating in a carefully curated MR data set, demonstrating the authors' commitment to data integrity.

Major concerns

1.Blinding and observer bias: Examiners performed fundoscopy unblinded to clinical presentation (Methods, page 4, lines 12-14). Knowledge of a child's coma severity or parasitaemia can subconsciously influence grading of subtle retinal findings. Without blinded re‐grading, we cannot rule out systematic inflation of diagnostic accuracy. Recommendation: A masked, independent re‐grading of stored retinal images by at least two readers (with κ statistics) is essential to confirm that the 1-5 haemorrhage threshold truly distinguishes CM from non-CM, rather than reflecting reviewer expectations.

2. Spectrum bias and generalisability: The cohort comprises only fatal CM cases (Fig 1, page 6), likely representing the most severe end of the disease spectrum. Retinal findings may be more pronounced in children who die, artificially boosting sensitivity and specificity. When applied to survivors or less severe cases, the refined criterion could underperform, leading to missed diagnoses or false reassurance. Recommendation: Either incorporate a non-fatal CM cohort or model the expected shift in test metrics (e.g. using the difference in median time to death between MR-/no sequestration and MR-/sequestration groups in Table 1, page 5) to demonstrate that the 1-5 haemorrhage cutoff will hold true across the full clinical spectrum.

3. Verification bias and missing data: 19 of 103 autopsied children lacked MR exams and were excluded (Fig 1, page 6), yet no comparison of baseline characteristics was presented. If excluded children systematically differed (for instance, younger age or lower parasitaemia), the retained sample may not be representative, skewing accuracy estimates. Recommendation: Compare age, parasitaemia, Blantyre Coma Score and other key variables between included and excluded patients (drawing from Table 1 data). Perform multiple imputation or sensitivity analyses to show that the 1-5 haemorrhage rule remains robust even when accounting for missing examinations.

4. Statistical precision and additional metrics: Table 3 omits positive and negative likelihood ratios (LR+ and LR-) and does not provide confidence intervals for the 12% specificity increase. Likelihood ratios translate test results into changes in disease probability at the bedside, directly informing clinical decisions. Confidence intervals for Δspecificity quantify uncertainty around the observed gain and guard against over-interpretation of chance findings. Recommendation: Calculate LR+ and LR- with 95% CIs (e.g., original PLR ~3.3 [2.0-5.7], recoded ~7.9 [3.1-19.0]) and report the CI for the specificity difference to demonstrate that the improvement is both clinically and statistically meaningful.

5. Borderline p-value interpretation and power: The specificity improvement yields p = 0.065 (Table 3, page 7), narrowly missing α = 0.05. Presenting a p value alone without context risks dichotomous thinking. A suggestive trend may nonetheless hold clinical importance, but without a post-hoc power analysis or confidence interval for Δspecificity, readers cannot assess whether the study was simply underpowered. Recommendation: Reframe p = 0.065 as a compelling trend, present the confidence interval for the specificity gain, and include a post-hoc power calculation (e.g., with n = 60, α = 0.05, the study has ≥80% power to detect a 15% specificity difference) to contextualize the finding's reliability.

6. Risk of over-fitting and need for external validation: The 1-5 haemorrhage threshold was both derived and tested within the same small subset (n = 60), raising the possibility of over-fitting. Without independent validation, there is a high risk that the cutoff works well only in this sample, limiting broader adoption and potentially leading to clinical misclassification elsewhere. Recommendation: Validate the refined threshold prospectively in an independent cohort—such as survivors from Beare et al. 2006 or a multi-centre study—with pre-specified performance targets and formal sample-size justification to confirm generalisability.

Statistical analysis and reporting

1. Threshold derivation transparency: Clarify whether the 1-5 haemorrhage cutoff was pre-specified based on prior data or emerged from exploratory analyses. If the latter, describe corrections for multiple testing to safeguard against Type I error.

2. ROC curve analysis: An ROC analysis of haemorrhage count (0 vs 1-5 vs >5) would yield an AUC, providing an objective measure of overall discrimination and reinforcing the choice of cutoff.

3. Handling of missing data: Specify the exact number and reasons for MR or sequestration data omissions; describe any imputation methods or decision rules applied to ensure complete transparency.

Presentation and clarity

1. Illustrative fundus images: Include high-quality examples of (a) 1-5 unilateral haemorrhages, (b) >5 bilateral haemorrhages, and (c) whitening/vessel changes, to guide clinicians in real-world grading.

2. Practical implementation tools: Provide a concise, step-by-step flowchart or checklist for non-ophthalmologist clinicians, enhancing consistent application of the refined criterion in busy, resource-limited wards.

3. Comprehensive integration of MR features: Discuss how the refined haemorrhage rule complements other MR signs—retinal whitening and vessel changes—in a combined diagnostic algorithm, maximizing overall accuracy.

Recommendation

Major revision. Addressing these methodological, statistical, and presentation enhancements will transform a promising retrospective finding into a robust, generalisable diagnostic tool, ultimately improving care and outcomes for children with suspected cerebral malaria in endemic regions.

---

* Please upload any figures associated with your paper as individual TIF or EPS files with 300dpi resolution at resubmission; please read our figure guidelines for more information on our requirements: http://journals.plos.org/plosmedicine/s/figures. While revising your submission, please upload your figure files to the PACE digital diagnostic tool, https://pacev2.apexcovantage.com/. PACE helps ensure that figures meet PLOS requirements. To use PACE, you must first register as a user. Then, login and navigate to the UPLOAD tab, where you will find detailed instructions on how to use the tool. If you encounter any issues or have any questions when using PACE, please email us at PLOSMedicine@plos.org.

* In you ethics statement, Please include IRB approval numbers.

* Please ensure that the study is reported according to the STARD guideline and include the completed STARD checklist as Supporting Information. When completing the checklist, please use section and paragraph numbers, rather than page numbers. Please add the following statement, or similar, to the Methods: "This study is reported as per STARD guideline (S1 Checklist)."

FIGURES AND TABLES

SUPPLEMENTARY MATERIAL

REFERENCES

DIAGNOSTIC STUDIES

* Please ensure that the study is reported according to the STARD guideline (https://www.equator-network.org/reporting-guidelines/stard/) and include the completed STARD checklist as Supporting Information. Please add the following statement, or similar, to the Methods: "This study is reported as per the Standards for Reporting of Diagnostic Accuracy (STARD) guideline (S1 Checklist)." When completing the checklist, please use section and paragraph numbers, rather than page numbers. 

* Please structure your Abstract according to STARD for Abstracts (https://www.equator-network.org/reporting-guidelines/stard-abstracts/).

* Please structure the Methods section using the following sub-headings: Study design, Participants, Test methods, Analysis.

* Please include a diagram to describe the flow of participants through the study (typically figure 1).

---

## [Decision Letter · Decision Letter 2]

1 Aug 2025

Dear Dr. Wilson,

Thank you very much for re-submitting your manuscript "Re-evaluating malarial retinopathy to improve its diagnostic accuracy in cerebral malaria" (PMEDICINE-D-25-01141R2) for review by PLOS Medicine. Please accept my apologies for the delay in providing you with a decision, due to attempts to obtain reviews from all the original reviewers.

I have discussed the paper with my colleagues and the academic editor and it was also seen again by 2 reviewers. I am pleased to say that provided the remaining editorial and production issues are dealt with we are planning to accept the paper for publication in the journal.

We look forward to receiving the revised manuscript by Aug 08 2025 11:59PM.   

Sincerely,

Suzanne De Bruijn, PhD

Associate Editor 

PLOS Medicine

plosmedicine.org

Requests from Editors:

* We kindly ask you to remove the power calculation (which was requested by R3, concern #5.), as post-hoc power calculations are considered bad statistical practice. Please accept my apologies for not catching this in the previous round of review, and the extra work that this has caused.

* Please address the remaining suggestions from Reviewer #1.

* Table 3: please specifcy the size of the confidence interval. is this 95%?

* Please remove the second period at the end of your Background section.

* We kindly ask you at this stage to include the URLs for Github and Zenodo.

* Are the MP_numbers in the supplemental data identifiers of the patients? Please review our guidance on patient confidentiality:

-It appears that you have included data that may breach patient confidentiality. Please edit your data set in accordance with patient consent, and remove any identifying data. For more information on how to share sensitive information please see the guidance here: https://journals.plos.org/plosmedicine/s/data-availability#loc-acceptable-data-access-restrictions

*can you clarify what supplemental information “32c2c9f8-42d8-446b-96fc-dfcb0b49de22” and “1c7a87b1-5594-4c02-a605-98c0175fd634” are?

GENERAL EDITORIAL REQUESTS

* Please confirm that your title complies with to PLOS Medicine's style. Your title must be nondeclarative and not a question. It should begin with main concept if possible. "Effect of" should be used only if causality can be inferred, i.e., for an RCT. Please place the study design ("A randomized controlled trial," "A retrospective study," "A modelling study," etc.) in the subtitle (ie, after a colon).

* Please confirm that your abstract complies with our requirements, including format (three sections: Background, Methods and Findings, and Conclusions) and providing all the information relevant to this study type https://journals.plos.org/plosmedicine/s/submission-guidelines#loc-abstract

* Please ensure that the Introduction ends with a clear description of the study question or hypothesis.

* Please ensure that all abbreviations are defined at first use throughout the text.

* Please confirm that all numbers presented in the abstract are present and identical to numbers presented in the main manuscript text.

* Please confirm that all original data are available in this manuscript. If this is not the case, please specify Whether these data can be obtained:

"

GENERAL

FUNDING STATEMENT

* The funding statement should include: specific grant numbers, initials of authors who received each award, URLs to sponsors’ websites. Also, please state whether any sponsors or funders (other than the named authors) played any role in study design, data collection and analysis, the decision to publish, or preparation of the manuscript. If they had no role in the research, include this sentence: “The funders had no role in study design, data collection and analysis, decision to publish, or preparation of the manuscript.”

* It appears that one or more study authors is affiliated with one or more of the agencies that funded the study. Thus, the statement “The funders had no role in study design, data collection and analysis, decision to publish, or preparation of the manuscript” does not apply. Please revise the Financial Disclosure accordingly, as in "[Author name] is [author's role] at [funding agency]. The funders had no other role in study design…..”

COMPETING INTERESTS STATEMENT

* All authors must declare their relevant competing interests per the PLOS policy, which can be seen here: https://journals.plos.org/plosmedicine/s/competing-interests For authors with ties to industry, please indicate whether any of the interests has a financial stake in the results of the current study.

ETHICS AND CONSENT

* At this time, please provide the name(s) of the institutional review board(s) that provided ethical approval, as well as the approval numbers.

FIGURES

* Please provide titles and legends for all figures and tables (including those in Supporting Information files). Please define all acronyms used in each figure or table in its corresponding legend.

DIAGNOSTIC TESTS

*Did your study have a prospective protocol or analysis plan? Please state this (either way) early in the Methods section.

* For all observational studies, in the manuscript text, please indicate: (1) the specific hypotheses you intended to test, (2) the analytical methods by which you planned to test them, (3) the analyses you actually performed, and (4) when reported analyses differ from those that were planned, transparent explanations for differences that affect the reliability of the study's results. If a reported analysis was performed based on an interesting but unanticipated pattern in the data, please be clear that the analysis was data-driven.

Comments from Reviewers:

Reviewer #1: The authors have addressed all my comments, and I am happy to recommend acceptance. I have included a few suggestions/notes below (numbers reference my original review items), but these are entirely optional. Well done and congratulations.

4. Suggest also using an alternate word for "significant" in the first sentence of the Discussion (line 228). 

6. Thank you - this is much clearer. I suggest also including a citation for the WHO criteria when first discussed at line 82.

20. My apologies - I should have flagged this as minor. I am OK with the authors' proposal. 

24. The revised Discussion is clear, cogent, and comprehensive. 

Reviewer #2: The authors have adequately addressed all my concerns in the edited manuscript.

---

## [Editor Report · Decision Letter 3]

21 Aug 2025

Dear Dr Wilson, 

On behalf of my colleagues and the Academic Editor, Lorenz von Seidlein, I am pleased to inform you that we have agreed to publish your manuscript "Re-evaluating malarial retinopathy to improve its diagnostic accuracy in pediatric cerebral malaria: a retrospective study" (PMEDICINE-D-25-01141R3) in PLOS Medicine.

Before your manuscript can be formally accepted we have 2 minor requests:

-for the 95% confidence intervals, could you please separate the lower and upper boundaries with comma's, instead of with hyphens?

-In table 3 you present P-values, could you please provide details of the statistical test used in the legend of this figure?

Furthermore, you will need to complete some formatting changes, which you will receive in a follow up email. Please be aware that it may take several days for you to receive this email; during this time no action is required by you. Once you have received these formatting requests, please note that your manuscript will not be scheduled for publication until you have made the required changes.

PRESS

Sincerely, 

Suzanne De Bruijn, PhD 

Associate Editor 

PLOS Medicine